# Is a definitive trial of prehospital continuous positive airway pressure versus standard oxygen therapy for acute respiratory failure indicated? The ACUTE pilot randomised controlled trial

Gordon Fuller ![ORCID],[1] Sam Keating,[1] Steve Goodacre ![ORCID],[1] Esther Herbert,[1] Gavin Perkins,[2] Andy Rosser,[3] Imogen Gunson,[3] Josh Miller,[3] Matthew Ward,[3] Mike Bradburn,[1] Praveen Thokala,[1] Tim Harris,[4] Maggie Marsh,[5] Alex Scott ![ORCID],[1] Cindy Cooper[6]

For numbered affiliations see end of article.

**Correspondence to**
Dr Gordon Fuller;
g.fuller@sheffield.ac.uk

## ABSTRACT

**Objectives** To determine the feasibility of a large-scale definitive multicentre trial of prehospital continuous positive airway pressure (CPAP) in acute respiratory failure.

**Design** A single-centre, open-label, individual patient randomised, controlled, external pilot trial.

**Setting** A single UK Ambulance Service, between August 2017 and July 2018.

**Participants** Adults with respiratory distress and peripheral oxygen saturations below British Thoracic Society target levels despite controlled oxygen treatment.

**Interventions** Patients were randomised to prehospital CPAP (O-Two system) versus standard oxygen therapy in a 1:1 ratio using simple randomisation.

**Primary and secondary outcome measures** Feasibility outcomes comprised recruitment rate, adherence to allocated treatment, retention and data completeness. The primary clinical outcome was 30-day mortality.

**Results** 77 patients were enrolled (target 120), including 7 cases with a diagnosis where CPAP could be ineffective or harmful. CPAP was fully delivered in 74% (target 75%). There were no major protocol violations. Full data were available for all key outcomes (targets ≥90%). Overall 30-day mortality was 27.3%. Of these deceased patients, 14/21 (68%) either did not have a respiratory condition or had ceiling of treatment decisions implemented excluding hospital non-invasive ventilation and critical care.

**Conclusions** Recruitment rate was below target and feasibility was not demonstrated. Limited compliance with CPAP, and difficulty in identifying patients who could benefit from CPAP, indicate that prehospital CPAP is unlikely to materially reduce mortality. A definitive effectiveness trial of CPAP is therefore not recommended.

**Trial registration number** ISRCTN12048261; Post-results.

## Strengths and limitations of this study

► This study provides important information regarding the feasibility of a large-scale definitive randomised controlled trial investigating prehospital continuous positive airway pressure in acute respiratory failure.

► Expert recommendations for pilot randomised controlled trials were followed.

► A pragmatic study design was used to maximise generalisability to real-life prehospital practice.

► A novel allocation concealment method using identical sealed boxes was used.

► Patients and clinicians were not blinded to the intervention.

## BACKGROUND

Acute respiratory failure (ARF) is a common medical emergency caused by cardiac or respiratory diseases, including heart failure, pneumonia, chronic obstructive pulmonary disease (COPD) and asthma.[1] The incidence of ARF has been estimated at 80 cases per 100 000 per year in the USA.[1] Although varying according to the underlying cause, the overall risk of death is high, with estimates of 30-day mortality ranging between 12% and 20%.[2] ARF has substantial health services costs, with patients often requiring prolonged hospital stays, ventilatory support and critical care admissions.[3 4] ARF is responsible for over 3 million National Health Service (NHS) bed days and hospital costs of £9.6 million per year in England.[5 6]

Current prehospital management of ARF in the NHS uses a standard management approach of controlled oxygen therapy,

BMJ

supplemented by specific ancillary treatments directed at the underlying disease.[7] This therapy is usually delivered by ambulance service personnel without physician support. Continuous positive airways pressure (CPAP) is widely used in hospital to treat ARF due to COPD, pneumonia, decompensated obstructive sleep apnoea, cardiogenic pulmonary oedema and chest wall trauma[8]; and it has been suggested that it may be more effective if delivered earlier, that is, en route to hospital.[5] The difficulties of prehospital diagnosis mean that prehospital CPAP is likely to be applied generally to all cases of ARF, rather than directed towards those due to a specific cause.

Existing research investigating prehospital CPAP is inconclusive. A recent evidence synthesis suggested that prehospital CPAP may reduce the risk of mortality in ARF compared with standard treatment, but noted that the primary studies were relatively small, heterogeneous, at risk of bias, and were provided by specialist services, so may not be applicable to routine prehospital practice.[5] These findings suggest that although prehospital CPAP is a promising and potentially efficacious therapy, a large pragmatic trial is needed to confirm effectiveness when implemented as part of routine practice across a clinically relevant population with ARF.

Prior to a large pragmatic trial and economic evaluation, it is first necessary to estimate the incidence of eligible patients to determine whether a trial would be feasible and cost effective. It is also important to determine whether prehospital CPAP can be delivered successfully in the context of the NHS ambulance services. For these reasons, a stand-alone feasibility study is necessary to estimate the incidence of eligible patients, test the feasibility and acceptability of potential definitive trial methods, and address important uncertainties, such as patient selection, delivery of the intervention and event rates.

## METHODS

### Study design, aims and objectives

The Ambulance CPAP: Use, Treatment effect and Economics (ACUTE) study was an open-label, individual randomised, pragmatic, parallel group, external pilot trial. It aimed to determine the feasibility of a definitive trial evaluating the effect of prehospital CPAP (intervention arm) compared with standard oxygen therapy (control arm) on 30-day mortality from ARF. The feasibility and effectiveness outcomes are listed in table 1. A protocol, with preplanned outcomes and analysis plan, has been previously published.[9] Health economic results will be reported separately.

### Setting and study population

Recruitment took place across four hubs covering 1.5 million people in an English ambulance service between 1 August 2017 and August 2018 (last follow-up completed 13 August 2018). Adults with respiratory distress and peripheral oxygen saturation below British Thoracic Society (BTS) target levels (88% for patients with COPD, 94% for other conditions) despite supplemental oxygen (titrated low flow oxygen for COPD, or titrated high flow oxygen in other conditions) were eligible.[10] Patients with pre-existing lack of capacity; or with limited potential to benefit from, or contraindications to CPAP, were excluded. Inclusion and exclusion

**Table 1** Ambulance CPAP: Use, Treatment effect and Economics trial objectives and outcomes

| Feasibility outcomes | Feasibility targets |
|---|---|
| 1. Rate of eligible patients per 100 000 population per year | Target 8 per 100 000 per year, that is, 120 across the 1.5 million population of the 4 hubs |
| 2. Proportion recruited and allocated to treatment appropriately | Proportion recruited in error and classified as major or minor non-compliances (target 0% and ≤10%). |
| 3. Adherence to allocated treatment | Adherence to the allocation schedule (target ≥90%) |
| | Adherence to treatment in the CPAP arm (target ≥75%) |
| 4. Retention and data completeness up to 30 days | Retention at 30 days (target ≥90%) |
| | Data completeness (target ≥90%) |

| Clinical outcome measures |
|---|
| 1. Proportion surviving to 30 days (primary outcome for definitive trial) |
| 2. Proportion undergoing endotracheal intubation by 30 days |
| 3. Proportion admitted to critical care at any point up to 30 days |
| 4. Mean and median length of hospital stay |
| 5. Change in Visual Analogue Scale dyspnoea score from presentation to immediately before ED arrival |
| 6. Mean change in quality of life, measured with EQ-5D-5L |
| 7. Key elements of postdischarge healthcare resource use up to 30 days |

CPAP, continuous positive airway pressure ; EQ-5D-5L, EuroQOL-5D-5L Value Sets.

## Box 1 Ambulance CPAP: Use, Treatment effect and Economics (ACUTE) trial eligibility criteria

**Inclusion criteria**
All of:
► Adults≥18 years
► Respiratory distress
► Hypoxiahypoxia:
  – Chronic obstructive pulmonary disease: Sats <88% despite low flow oxygen
  – Other conditions: Sats <94% despite high flow oxygen.

**Exclusion criteria**
Any of:
1. Hospital continuous positive airways pressure (CPAP) treatment available within 15 min of eligibility assessment
2. Age <18 years
3. Known to have terminal illness
4. Known pre-existing lack of capacity (confirmed by relatives, carers or documentary evidence, such as Lasting Power of Attorney)
5. Documented not for resuscitation status
6. Acutely incapacitated patients with known valid advanced directive declining non-invasive ventilation or participation in research
7. The patient has an oxygen alert card*
8. Anticipated inability to apply CPAP (eg, facial deformity)
9. Respiratory failure due to chest trauma
10. Contraindication to CPAP (suspected pneumothorax, respiratory arrest, epistaxis, vomiting, hypotension)
11. Previous enrolment in the ACUTE trial
12. Pregnancy
13. Patients unable to communicate with ambulance service clinicians

*Patient-held card warning against use of high flow oxygen. Provided to patients with a history of previous hypercapnic acidosis with a $PaO_2$ >10.0 kPa—indicating that oxygen may have worsened the hypercapnia.

criteria are detailed in box 1 and were based on clinician judgement at the scene of incident.

### Randomisation

Enrolled patients were individually allocated to CPAP or standard oxygen therapy in a 1:1 ratio using simple randomisation. The randomisation sequence was computer generated by an independent statistician not directly involved in the conduct of the trial. The allocation schedule was held centrally on a password-protected, access-restricted network drive. The trial statistician did not have access to the randomisation sequence until after data lock.

### Participant enrolment and consent

Ambulance service clinicians (paramedics and ambulance technicians) volunteering to participate in the ACUTE study and trained in trial procedures identified potential participants with ARF when attending emergency 999 ambulance calls. In patients judged to have capacity, verbal consent was obtained for participation. Eligible patients lacking mental capacity were enrolled in the trial without consent if determined to be in their best interests. In all cases, a research paramedic reviewed the participant in hospital as soon as possible after enrolment, obtaining consent for further data collection and participation in the trial according to the provisions of the Mental Capacity Act 2005.[11 12] In the contingency that consent was declined, prehospital data were retained and anonymised 30-day mortality data collected. If a patient died before approach for written consent, ethical approval allowed for collection of anonymised prehospital, hospital and 30-day mortality data.

### Allocation concealment and blinding

CPAP devices and high concentration oxygen therapy masks were packaged in identically appearing, numbered, shrink wrapped, tamper proof sealed, trial equipment boxes. Equipment boxes were independently assembled in an audited process in accordance with the randomisation sequence at Sheffield Clinical Trials Research Unit. Boxes were supplied to participating ambulance hubs, and held unordered by number, in a secure designated storage area. Boxes were subsequently signed in and out for each shift by participating clinicians. Research paramedics audited the location and condition of all boxes, and adherence to the allocation schedule on a weekly basis. Due to the physical differences between the CPAP device and standard oxygen mask, it was not possible to subsequently blind patients, ambulance service clinicians, hospital clinicians or outcome assessors to the treatment arms.[9]

### Trial treatments

Immediately after enrolment, paramedics opened the trial equipment box and provided treatment according to whether a CPAP device or high concentration oxygen mask was supplied. Patients in the intervention arm were treated with the O-Two unit (O-Two Medical Technologies, Ontario, Canada; supplied by SP Services (UK), Telford, UK), a lightweight, portable, open, single use, low flow CPAP system.[7] CPAP was started at 5 cm $H_2O$ and then incrementally increased by 1 cm $H_2O$ every 2–5 min to a maximum of 15 cm $H_2O$. Patients in the control arm received standard oxygen therapy using nasal cannula, an air entrainment Venturi mask, a simple face mask or a non-rebreathing reservoir face mask. The choice of flow rate and oxygen delivery device was titrated by ambulance service clinicians according to the patient's condition, and peripheral oxygen saturation levels.

Treatment in both arms was targeted to BTS guidelines for peripheral oxygen saturations.[8] Target peripheral oxygen saturations were 88%–92% for patients with known/suspected COPD and 94%–98% for patients with other suspected causes of ARF. Ancillary condition-specific treatments were administered in both trial arms according to Joint Royal Colleges Ambulance Liaison Committee Clinical Practice Guidelines.[7] Subsequent hospital management, including emergency department CPAP or non-invasive ventilation (NIV), was at the discretion of the hospital clinician.

## Data collection and safety reporting

Electronic patient records and computer-aided dispatch data were screened by research paramedics to identify potentially eligible, but unenrolled, patients. Routinely collected prehospital data (eg, timings, vital signs) were collated from participants' electronic records by research paramedics. Additional trial specific information was collected using a trial form completed by recruiting ambulance service clinicians and contained within each equipment box. At 30 days, research paramedics reviewed the hospital records to collate details of subsequent progress, treatments and vital status at 30 days. Quality of life and postdischarge resource use was assessed by questionnaire at 30 days following enrolment, either in person if still in hospital, or by telephone or post if discharged. Adverse health changes in participants were defined, monitored, recorded and reported according to UK Health Research Authority guidance.[13]

## Sample size and statistical analyses

The baseline characteristics of enrolled patients are reported descriptively for the whole trial population, and separately per treatment arm. Feasibility outcomes are reported descriptively for the whole trial population, together with their 95% CI. Summary estimates of relative and absolute effectiveness outcomes are presented, overall and stratified by treatment arm, with 95% CIs. An intention-to-treat analysis was used, using either a full analysis set, or a complete case analysis for endpoints with missing outcome data.[14]

The ACUTE feasibility study aimed to recruit 120 patients over 12 months, the minimum number recommended to estimate binary parameters for the sample size calculation of the full trial.[15] Mortality under standard care was estimated at 12% and for the full trial a 5% absolute reduction was postulated (ie, to 7%) in the intervention arm.[1 3 4] This sample size would allow estimation of mortality to within an SE of 2.7% for use in the sample size calculation for any definitive trial, and feasibility outcomes to be estimated with a precision of <5%. Achieving this target would also indicate a recruitment rate sufficient to deliver the sample size for a definitive trial, and an incidence of ARF necessary to deliver a potentially cost-effective prehospital CPAP service.

All analyses were conducted in the R statistical package (R Foundation for Statistical Computing, Vienna, Austria) in accordance with a prespecified statistical analysis plan and Consolidated Standards of Reporting Trials principles. Following review of blinded outcome data during trial management group (TMG) meetings, an unexpectedly high overall mortality rate was noted. A post-hoc descriptive analysis of the deceased patients was therefore conducted.

## Public and patient involvement

The public and patients were fully involved in the ACUTE study from conception to dissemination. The research proposal was developed in partnership with a service user

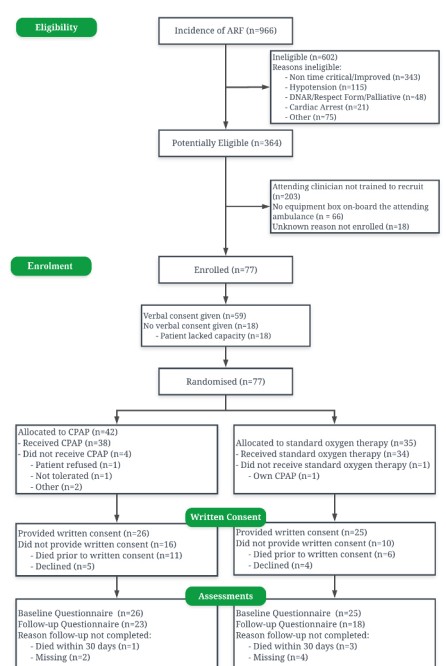

**Figure 1** Consolidated Standards of Reporting Trials diagram of participants' flow through the Ambulance CPAP: Use, Treatment effect and Economics pilot trial. ARF, acute respiratory failure; CPAP, continuous positive airways pressure; DNAR, Do not attempt resuscitation.

coapplicant and patient advisory groups. A service user advisory group was enlisted for collaboration throughout the project, and a lay coapplicant provided advice on trial matters, attended TMG meetings, supported study management and contributed to the study report and interpretation of result

## RESULTS
### Recruitment and consent

Over the recruitment period, 364 patients with ARF meeting ACUTE trial eligibility criteria presented from the 1.5 million population of four participating ambulance service hubs. One hundred sixty-one of those patients (44.2%) were attended by ACUTE trained ambulance service clinicians and could potentially have been recruited. Of these 161, 77 (47.8%) were enrolled in the trial. Slightly more participants were randomised to the CPAP intervention arm (42 cases), than to the standard oxygen control arm (35 cases). Consent for further data collection was declined by 9 patients (11.7%) and 17 patients (22.1%) died prior to research paramedic approach for consent. Figure 1 presents the participant flow for the trial.

### Baseline characteristics

The trial population was relatively elderly (median age 71 years), predominantly male (62.3%) and severely unwell (median Visual Analogue Scale (VAS) breathlessness score 9/10, median pulse 115 beats/min, median respiratory rate 34 breaths/min and median initial peripheral

**Table 2** Baseline characteristics of Ambulance CPAP: Use, Treatment effect and Economics trial participants by arm

| Baseline variable | Descriptive statistic | CPAP N=42 | Standard oxygen therapy N=35 | Total N=77 |
|---|---|---|---|---|
| Age | Median (IQR) | 70.0 (60.5 to 76.8) | 73.0 (65.0 to 77.0) | 71.0 (62.0 to 77.0) |
| Sex | Male | 27 (64.3%) | 21 (60.0%) | 48 (62.3%) |
| Hospital ARF diagnosis* | n= | 36 | 30 | 66 |
| | Asthma | 1 (2.8%) | 1 (2.9%) | 2 (3.0%) |
| | COPD | 10 (27.8%) | 11 (36.7%) | 21 (31.8%) |
| | Heart Failure | 4 (11.1%) | 2 (6.7%) | 6 (9.1%) |
| | LRTI | 17 (47.2%) | 11 (36.7%) | 28 (42.4%) |
| | Pulmonary embolism | 1 (2.8%) | 0 (0.0%) | 1 (1.5%) |
| | Other | 3 (8.3%) | 4 (13.3%) | 7 (10.8%) |
| Breathlessness at enrolment (VAS 0–10) | n= | 41 | 35 | 76 |
| | Median (IQR) | 9.0 (8.0 to 10.0) | 9.0 (8.0 to 9.5) | 9.0 (8.0 to 10.0) |
| First systolic blood pressure (mm Hg) | n= | 40 | 30 | 70 |
| | Median (IQR) | 136.0 (115.2 to 150.5) | 126.5 (112.0 to 152.0) | 134.5 (112.2 to 152.0) |
| First Glasgow Coma Score | n= | 42 | 35 | 77 |
| | Median (IQR) | 15.0 (14.0 to 15.0) | 15 (14.5 to 15.0) | 15 (14.0 to 15.0) |
| First oxygen saturations (%) | n= | 41 | 35 | 76 |
| | Median (IQR) | 78 (74.0 to 85.0) | 82 (75.5 to 86.0) | 78.5 (74.8 to 86.0) |
| First pulse rate (beats/min) | n= | 42 | 33 | 75 |
| | Median (IQR) | 117.0 (105.0 to 125.8) | 111 (92.0 to 121.0) | 115 (100.0 to 124.0) |
| First respiratory rate (breaths/min) | n= | 42 | 35 | 77 |
| | Median (IQR) | 36.0 (30.5 to 40.0) | 32 (24.0 to 40.0) | 34 (28.0 to 40.0) |
| Duration between arrival at scene and departure to hospital (min) | Median (IQR) | 43.0 (34.0 to 49.8) | 36.0 (32.5 to 46.5) | 40.00 (34.0 to 49.0) |
| Duration between leaving the scene and arriving at hospital (min) | Median (IQR) | 13.0 (9.00 to 18.75) | 15.0 (10.0 to 20.5) | 13.0 (10.0 to 20.0) |

'N' refers to the total sample. 'n' refers to the subgroup numbers.
*Consent was declined for data collection in nine cases, clinical records were unavailable in two cases, and in one case there was no clear underlying diagnosis apparent in the notes.
ARF, acute respiratory failure; COPD, chronic obstructive pulmonary disease; LRTI, lower respiratory tract infection; VAS, Visual Analogue Scale.

oxygen saturations of 78.5%). The most common final hospital diagnoses were COPD (n=21/65, 32.3%) and lower respiratory tract infection (n=28/65, 43.1%). A minority of cases (n=4/65, 6.2%) had a non-respiratory primary hospital diagnosis comprising abdominal aortic aneurysm, myocardial infarction, sepsis (not further specified) and liver failure (ascites). Prerandomisation characteristics were similar across trial arms as summarised in table 2.

### Feasibility outcomes
Feasibility outcomes, compared with the prespecified target, are summarised in table 3. The sample size of 77 enrolled patients resulted in a recruitment rate of 5.1 (95% CI 4.1 to 6.4) per 100 000 population per year,

falling short of the feasibility target of 8 per 100 000 population per year (ie, 120 patients).

All patients were recruited appropriately with no major or minor protocol non-compliances (targets of ≤10% and 0%, respectively). There was full adherence to the allocation schedule (feasibility target ≥90%). Treatment with CPAP was attempted in all patients enrolled in the intervention arm. All patients received appropriate standard oxygen management in the control arm, although one patient was enrolled who used their own CPAP machine.

CPAP was fully delivered as planned (ie, administered until hospital arrival, or discontinued due to patient improvement after successful treatment) in 73.8% (31/42, target 75%) of intervention arm patients. CPAP

**Table 3** Summary of feasibility results

| Feasibility outcome | Target | Result |
|---|---|---|
| Recruitment rate | 8 per 100 000 population per year (ie, 120 patients recruited) | ► 5.1 (95% CI 4.1 to 6.4) per 100 000 population per year<br><br>► 77 enrolled patients. |
| Major and minor non-compliances | 0% and ≤10% | ► 0%—major non-compliances<br><br>1.3%—minor non-compliance |
| Adherence to the allocation schedule | Target ≥90% | ► 100% adherence to allocation schedule |
| Adherence to treatment in the CPAP arm | Target ≥75% | ► 74%—CPAP fully delivered as planned |
| Retention at 30 days | Target ≥90% | ► 100%—follow-up for all feasibility endpoints and 30-day mortality |
| Data completeness | Target ≥90% | Data completeness for outcomes:<br>► 100%—feasibility outcomes<br>► 100%–30-day mortality<br>► 81%–30-day intubation<br>► 84%—admission to critical care<br>► 99%—clinician assessed breathlessness<br>► 86%—length of hospital stay<br>► 85%—baseline EQ-5D-5L*<br>► 71%–30-day EQ-5D-5L*<br>► 73%–30-day resource use* |

*Of alive patients.
CPAP, continuous positive airways pressure; EQ-5D-5L, EuroQOL-5D-5L Value Sets.

was commenced in 90.5% of patients (38/42), with two patients refusing to wear the mask, one patient spontaneously improving and a fourth patient having a cardiac arrest prior to commencement. Of patients commencing CPAP, 31 (81.6%) continued with CPAP until they arrived at hospital: 6 did not tolerate CPAP and the remaining patient (with a final diagnosis of pulmonary fibrosis) was transferred to a standard oxygen non-rebreather mask due to non-improvement.

Full data were available for key outcomes, including all feasibility endpoints and vital status at 30 days, compared with the feasibility targets of ≥90% retention at 30 days and ≥90% data completeness. A small amount (<1%) of prehospital data describing baseline patient characteristics was not available due to missing values. A larger proportion of hospital data was missing secondary to lack of consent for collection (n=9, 11.7%), absence of local research approvals to access hospital data (n=2, 2.6%), or unclear or absent information in the hospital clinical records (differing across variables, ranging from n=0, 0%, for hospital length of stay to n=25, 32.5%, for emergency

department management). Of patients alive at 30 days (n=56), 30-day follow-up questionnaires examining quality of life and postdischarge health resource use were fully completed by 40 patients (71.4%).

### Effectiveness and safety outcomes
Overall mortality of the study population was higher than expected with 27.3% (n=21/77) patients dying by 30 days. 28.6% (n=12/42) patients died by 30 days in the CPAP arm and 25.7% (n=9/35) in the standard oxygen arm (expected 12%). The absolute risk difference of mortality in CPAP group compared with standard oxygen therapy was 2.9% (95% CI –19.7% to 25.4%). An unplanned descriptive analysis of the deceased patients was undertaken to explore the circumstances for the unexpectedly high mortality rate. Data were not available for two cases due to lack of research approvals to access data, and two patients died from non-cardiorespiratory conditions not amenable to NIV (ruptured abdominal aneurysm and liver failure). Of the remaining 17 cases, 6 patients (35.3%) received hospital NIV (n=5/17,

Table 4  Summary of effectiveness outcomes

| Effectiveness outcome | CPAP N=42 | Standard oxygen therapy N=35 | Total N=77 | Absolute risk difference (95% CI) |
|---|---|---|---|---|
| 30-day mortality | n=42 12 −28.60% | n=35 9 −25.70% | n=77 21 −27.30% | 2.90% (−19.7 to 25.4) |
| Intubated | n=33 2 −6.10% | n=29 1 −3.40% | n=62 3 −4.80% | 2.60% (−10.5 to 15.7) |
| Admission to critical care | n=35 4 −11.40% | n=30 2 −6.50% | n=65 6 −9.20% | 4.80% (−12.1 to 21.7) |
| | | | | Median of the difference (95% CI) |
| Median length of stay (days, IQR) | n=22 10 (6.5 to 12) | n=22 7 (5 to 9.8) | n=44 8 (5.5 to 11.2) | 3 (−0.00 to 6.00) |
| Change in patient reported breathlessness over prehospital interval (VAS, IQR) | n=18 −3 (−4 to −2) | n=18 −2 (−4 to −1) | n=36 −2.5 (−4 to −1) | −2 (−3.00 to 0.00) |
| Change in clinician assessed breathlessness over prehospital interval (VAS, IQR) | n=41 −3 (−5 to −1) | n=35 −2 (−3.5 to 1) | n=76 −2 (−4 to −1) | −1 (−2 to 0) |
| 30-day EQ-5D-5L (IQR) | n=22 0.82 (0.58 to 0.95) | n=18 0.73 (0.43 to 0.89) | n=40 0.76 (0.48 to 0.92) | 0.08 (−0.6 to 0.26) |
| Median change in EQ-5D-5L (IQR) | n=22 0.09 (−0.01 to 0.16) | n=18 0.1 (−0.06 to 0.19) | n=40 0.09 (−0.02 to 0.18) | 0.0 (−0.12 to 0.16) |

'N' refers to the total sample. 'n' refers to the subgroup numbers.
EQ-5D-5L, EuroQOL-5D-5L value sets; VAS, Visual Analogue Scale.

29.4%) or mechanical ventilation (n=1, 5.9%). The other 11 patients had hospital ceiling of treatment decisions excluding hospital NIV and critical care management (11/17, 64/7%). Other secondary effectiveness outcomes are summarised in table 4, and were similar across trial arms. Postdischarge resource use is summarised in online supplementary file 1.

In total, 39 expected related serious adverse events (SAE) occurred in 34 patients (45.2% for the CPAP arm, 42.9% for the standard oxygen therapy arm). The majority were deaths (20/39, 51.2%) or readmission to hospital within 30 days (13/39, 33.3%). There were no unexpected related SAEs. Two patients (one intervention arm patient not receiving CPAP and one control arm patient) were categorised with related expected SAEs following diagnosis with pneumothoraces requiring intercostal drainage after hospital admission. There were no

significant SAEs attributable to CPAP therapy. Adverse events are detailed in online supplementary file 2.

## DISCUSSION
### Summary of results
Over 12 months, 77 patients were enrolled, below the recruitment target of 120 participants. CPAP was fully delivered as planned in 74% of intervention arm patients (target 75%). There were no major protocol violations. Full data were available for key outcomes, including all feasibility endpoints and vital status at 30 days. Mortality was higher than expected, with 27.3% of patients dying by 30 days. Of the deceased patients, 68% either did not have a respiratory condition or had explicit or implicit ceiling of treatment decisions excluding hospital NIV or critical care. Two patients, neither receiving CPAP,

were diagnosed with a pneumothorax in the emergency department and were reported as expected related SAEs.

## Interpretation

These findings suggest that pilot study methods could be used for a future definitive trial. Patients were appropriately recruited using a deferred consent model, there were no protocol violations, adherence to the allocation schedule was complete and full data were available for key outcomes. The ACUTE pilot trial demonstrated a recruitment rate that was below the target rate considered a priori necessary to deliver a definitive pragmatic trial. However, the majority of potentially eligible cases presented to non-trial trained staff and it is therefore possible that recruitment could be improved if a full trial mandated participation of all ambulance service personnel.

External pilot trials are not designed or powered to generate estimates of clinical effect that should be used for decision-making.[16–18] We therefore draw no conclusions from comparisons of outcomes between CPAP and standard care. However, a number of findings from the ACUTE pilot trial can be used to inform a judgement regarding whether it would be plausible for a large trial to detect an effect from CPAP on mortality and therefore whether a definitive trial might be worthwhile.

First, it is apparent that prehospital identification of patients with the potential to benefit from CPAP may be challenging. A small, but significant minority of cases were ultimately diagnosed with conditions where CPAP could not conceivably be beneficial.[8 19] It is also concerning that pneumothoraces requiring intercostal drainage were detected in two cases. Although neither patient received prehospital CPAP, the potential for iatrogenic harm is conspicuous.

Second, delivery of CPAP was relatively limited, with 74% of intervention arm patients continuing treatment to hospital as planned. Although CPAP may be efficacious, the potential to demonstrate effectiveness would be restricted by lack of treatment compliance.

Third, a key rationale for the implementation of prehospital CPAP is that earlier instigation of treatment will improve outcomes over and above the availability of hospital NIV.[5] In the pilot trial, relatively short on-scene times and conveyance times were recorded (median of 40 min and 13 min, respectively). The potential time advantage from prehospital administration of CPAP may be too small to produce meaningful benefit.

Finally, illness severity was much higher than anticipated. The increased overall 30-day mortality risk of 27.3% might initially appear to offer a definitive trial a greater opportunity to detect a clinically relevant survival benefit. However, many enrolled patients had apparent treatment limitation decisions for ward-level hospital care only, indicating a study population with a high prevalence of end-stage cardiorespiratory disease, multiple severe comorbidities or very poor premorbid performance status where CPAP treatment might be futile and overly burdensome. However, it should be noted that CPAP could also affect important patient-centred outcomes such as symptom relief, which might be detected by differences in dyspnoea VAS scores.

In summary, although the higher than expected mortality rate might suggest increased potential to detect an absolute difference in mortality, the challenges of providing prehospital CPAP, and characteristics of the patients who would benefit from CPAP, suggest limited potential to improve survival. It therefore appears unlikely that a trial powered to detect a plausible effect size could be designed.

## Limitations

The purpose of a feasibility study is to determine if a large-scale trial can be performed.[18] Therefore, challenges that might be interpreted as weaknesses when appraising a definitive RCT (such as low rates of recruitment and adherence) actually represent important learning points in the feasibility setting. Modification of eligibility criteria to specify the presence of a primary cardiorespiratory diagnosis as an inclusion criteria, and exclusion of patients with home CPAP machines, could select a more appropriate study population. The use of retrospective case note review for hospital data collection improved the efficiency of data collection, but is associated with well-recognised limitations that could have resulted in inaccurate measurement of study endpoints. Furthermore, collection of more detailed data on NIV treatment in the emergency department would be beneficial in any future trial.

## Generalisability

Trial eligibility criteria were broad, excluding only patients where CPAP was contraindicated (eg, vomiting), where entry would likely not be appropriate (eg, do not resuscitate status, pre-existing loss of capacity), or where verbal consent was not possible (eg, language barrier). The clinical findings reported herein should therefore have strong generalisability to patients presenting with ARF in NHS ambulance services. External validity to other settings, with different demographics or prehospital systems, is less clear. For example, international emergency medical services may have shorter on-scene times, use CPAP in less severe ARF, or use prehospital physicians with ultrasound skills to detect pneumothorax before CPAP application. Furthermore, a novel CPAP device was used. Although there are considerable advantages to the studied O-Two unit including small size, low cost and simplicity, there may also be limitations compared with other more complex prehospital CPAP systems, for example, $FiO_2$ and CPAP level are jointly determined by the oxygen flow rate. Consequently, efficacy could differ with other methods of delivering prehospital CPAP.

## Comparison to previous literature

A series of recent observational studies have also demonstrated that CPAP can be implemented by Emergency

Medical Services (EMS).[20–24] These studies are consistent with the ACUTE experience of submaximal CPAP adherence and difficult prehospital diagnosis, including treatment of patients with pneumothoraces. A recent systematic review identified 10 trials and quasi-randomised studies comparing prehospital NIV (including CPAP) with standard oxygen therapy.[5] Network meta-analysis suggested that prehospital CPAP is an effective treatment for ARF, with evidence that it reduces mortality (OR 0.41; 95% CrI 0.20 to 0.77) and intubation rate (0.32; 95% CrI 0.17 to 0.62) compared with standard care.[2] However, some included studies were at risk of selection bias from lack of allocation concealment and information bias, secondary to unblinded outcome assessment. Furthermore, the meta-analysis findings have doubtful external validity to prehospital practice in many settings. Only one trial included patients with undifferentiated respiratory failure, and the methods used to deliver prehospital CPAP (physician or paramedics with online physician support) are not routine in many prehospital care systems, reflected in higher levels of CPAP compliance.[5]

## CONCLUSIONS

Pilot trial recruitment rate was below the target rate. Limited compliance with CPAP, and a study population including patients who could not benefit from CPAP, suggest that a clinically significant effect size is not plausible. A definitive effectiveness trial of CPAP is therefore not recommended. These findings also argue against routine implementation of CPAP into many ambulance services, but would not preclude a CPAP service provided by clinicians with extended training (eg, prehospital physicians) where advanced clinical skills might allow selective targeting of treatment to an appropriate subgroup of patients.

**Author affiliations**
[1] School of Health and Related Research, The University of Sheffield, Sheffield, UK
[2] Clinical Trials Unit, University of Warwick, Coventry, UK
[3] West Midlands Ambulance Service NHS Foundation Trust, Brierley Hill, UK
[4] School of Medicine and Dentistry, Blizard Institute, Barts and The London School of Medicine and Dentistry, London, UK
[5] Sheffield Emergency Care Forum, Sheffield, UK
[6] Clinical Trials Research Unit, University of Sheffield, Sheffield, UK

**Contributors** The following drafted the manuscript: GF (Chief Investigator), SG (Co-Chief Investigator), SK (Trial Manager), EH (Statistician), MB (Coapplicant, statistician), PT (Coapplicant, health economist), JM (Research paramedic), MM (Coapplicant, PPI representative). The following conceived of, or designed, the work: GF, SG, SM, PT, MM, MB, GP (Coapplicant), MW (Coapplicant), TH (Coapplicant), MM, AS (Qualitative researcher), CC (Coapplicant, CTRU Director). The following were involved in the acquisition of data for the work: AR (Lead Research Paramedic), MW (Coapplicant), IG (Research paramedic) and JM (Research paramedic). The following were involved in the analysis and the interpretation of data for the work: GF, SG, SK, EH, MB, PT, JM, IG, MM and AS. The following contributed to the management and conduct of the trial: SK, SG (Trial support officer), AR, MW. All authors revised the work critically for important intellectual content and were involved in the final approval of the version to be published.

**Funding** This work was supported by the National Institute for Health Research's Health Technology Assessment Programme, grant number HTA15/08/40. The

funder, and CPAP manufacturer and supplier, were not involved in study design, conduct or publication.

**Competing interests** SG is Deputy Director of the NIHR HTA Programme, Chair of the NIHR HTA Commissioning Board and member of the NIHR HTA Funding Strategy Group. GP is an NIHR Senior Investigator and member of the Programme Grants for Applied Research Board. CC is a member of the NIHR Clinical Trials Unit Standing Advisory Committee and of the UK Clinical Research Collaboration Registered Clinical Trials Unit Network Executive Group. All authors have completed the unified competing interest form at http://www.icmje.org/coi_disclosure.pdf (available on request from the corresponding author).

**Patient and public involvement** Patients and/or the public were involved in the design, or conduct, or reporting, or dissemination plans of this research. Refer to the Methods section for further details.

**Patient consent for publication** Not required.

**Ethics approval** The trial received ethical approval from the NHS Leeds East Research Ethics Committee (Reference: 16/YH/0406). A protocol amendment from permuted block to simple randomisation was approved prior to recruitment (16 February 2017, REC reference: 16/YH/0406).

**Provenance and peer review** Not commissioned; externally peer reviewed.

**Data availability statement** Data are available upon reasonable request. All data requests should be submitted to the corresponding author for consideration. Access to anonymised data may be granted following review.

**ORCID iDs**
Gordon Fuller http://orcid.org/0000-0001-8532-3500
Steve Goodacre http://orcid.org/0000-0003-0803-8444
Alex Scott http://orcid.org/0000-0001-7426-7099

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
