## [Reviewer comments · BMJ Open]

ARTICLE DETAILS

TITLE (PROVISIONAL)	Is a Definitive Trial of Prehospital Continuous Positive Airway Pressure versus standard oxygen therapy for Acute Respiratory Failure Indicated? The ACUTE pilot randomised controlled trial
AUTHORS	Fuller, Gordon; Keating, Sam; Goodacre, Steve; Herbert, Esther; Perkins, Gavin; Rosser, Andy; Gunson, Imogen; Miller, Josh; ward, matthew; Bradburn, Mike; Thokala, Praveen; Harris, Tim; Marsh, Maggie; Scott, Alex; Cooper, Cindy

VERSION 1 – REVIEW

REVIEWER	Vibe Maria Laden Nielsen, MD Centre for Prehospital and Emergency Research, Aalborg University & Aalborg University Hospital, Denmark
REVIEW RETURNED	19-Dec-2019

GENERAL COMMENTS	Major methodological considerations Eligibility: Please beware and comment on the fact that this study's inclusion criteria states that patients with hypoxia despite oxygen treatment are eligible. This is not consistent with the inclusion criteria in 4 of the 5 referred studies in your discussion, where patients are included if hypoxia is present before ANY treatment. Is it possible to add details to the eligibility criteria regarding prior oxygen treatment, e.g. did you define "hypoxia despite oxygen treatment"? For how long did ambulance clinicians wait for improvement with only oxygen treatment? The eligibility criteria arguably result in a patient population with more severe illness stages in the present study and challenge comparison between this and other studies (also reflected in the high mortality rate). Interventions: Intervention arm: did you collect data on CPAP pressures (cm H2O) applied? How many patients were treated with 5, 8, 10, 15, 20 or 25 cm H2O respectively? In relation to the CONSORT guideline item 11b on similarity of interventions; maybe it would be worth to mention that both treatment groups were treated with oxygen of varying FiO2? Truth be told, the intervention is not only CPAP but also oxygen treatment and we do not know the FiO2 values in either of the groups. Had we known, oxygen treatment could have been adjusted for when comparing hard outcomes. And how many of the patients in each group did reach BTS target levels of peripheral oxygen SATs after CPAP or standard treatment? Is it possible to comment on or show if any patients receiving CPAP deteriorated with worsened SATs, RRs or VAS dyspnoea scores upon arrival at the emergency department?
--

Confounding:

Do you have any data on CPAP treatment in the emergency department? This is a substantial confounder for outcomes in both groups. I noticed in the published protocol that "... time to receiving hospital NIV, if provided" would be collected? Is it possible to comment on the general use of emergency department CPAP in England - is it implemented in all hospitals or does it rely on local guidelines?

Results:

No 95% confidence intervals are reported for risk differences in mortality and other outcomes between treatment groups. Having read the references 17-19 provided by the authors, to my understanding, there is no consensus about the need to report 95% confidence intervals on treatment outcomes from a pilot study. According to Lee et al., 95% CIs give an estimation of the treatment effect size i.e. what to expect from a definitive study (please read page 2 in the research paper attached (Lee)). I kindly suggest the authors to reconsider reporting of 95% CIs, also in the abstract. Especially since the conclusion is that CPAP is unlikely to materially reduce mortality. And it has also come to my attention reading the published study protocol; "... effectiveness outcomes will be reported descriptively for the whole trial population, and each trial arm, together with their 95% confidence intervals."

I also suggest adding your result of patient VAS breathing score of -3 after CPAP treatment to the discussion. Yes, CPAP might be futile, if there is no effect on mortality, but if the patients report symptom relief, the treatment is still valued by the patient and symptom relief is a valid treatment indication as well (provided the treatment is deemed safe of course).

Adherence:

Adherence to CPAP treatment rate was very close to target value (75%). In a definitive RCT, the design could have pragmatic treatment arms "prehospital CPAP for as long as the patient can tolerate" and "standard care". I kindly draw the authors' attention to other recent CPAP studies, where adherence to CPAP treatment was 88% in a cohort of >150 ARF patients and 83,5% in a cohort of >300 ARF patients receiving prehospital CPAP treatment (Nielsen et al., Bledsoe et al., I have attached these two research papers for your inspiration).

Conclusion:

One might challenge your conclusion that a definitive effectiveness trial of prehospital CPAP is not recommendable. Recruitment rate could be raised by training more staff and adherence was very close to target value.

A set of revised eligibility criteria including less severely ill patients, i.e. hypoxia before any oxygen treatment, similar FiO₂s in both treatment groups and information on continued CPAP in the ED or not may strengthen a randomised study design of CPAP effectiveness.

On the other hand, one might argue that the apparent difference in 30 day mortality favouring standard care should be a reason for not conducting a large scale RCT due to ethical problems of potential harm.

In my opinion, there is still controversy about this treatment option and a large scale RCT would be of great value. I fully agree with your last conclusion and I congratulate you on finishing and

	publishing this most impressive and comprehensive work. Minor comments P1L3: I recommend no abbreviations in the title and suggest CPAP to be spelled out as Continuous Positive Airway Pressure. I also suggest including "versus standard oxygen therapy" to clarify in the title what both of the treatment arms of the RCT include. P2L18: Contributions: It might be simpler if each person's title e.g. "Chief Investigator" was noted only following the first time his or her name is mentioned. Then in the following paragraphs, only initials are used and no repetition of titles. Consider moving "The following drafted the manuscript" down to P3L2. P4L13: Consider to delete specific centre name (according to BMJ Open submission guidelines, the name of a specific centre is irrelevant in the abstract). P4L32: It would be more precise if it was clarified in the abstract, that both the reported 30 day mortality and the following phrase are outcomes for the total study population. You may consider including results regarding any adverse events from CPAP rather than results from the post-hoc analysis of deceased patients, since the primary study objective concerns feasibility. P4L40: Consider concluding on the decision not to extend to a large-scale RCT in the abstract. P4L44: Please specify the name of trial registry. P4L46: BMJ Open prefers a standard sentence: "This work was supported by [name of funder] grant number [xxx]". P5L11: This section should not comment on the study's results but solely on the methods (please refer to BMJ Open submission guidelines for authors). P6L9: Is this the incidence rate in England? P6L25: Please specify the indications/diagnoses that CPAP is used for in hospital. P6L26: "prehospital diagnostics"? P7L19: An idea could be to cite your own protocol article here, in which you describe the EMS setting in detail. In your EMS setting, is it possible for 'ambulance clinicians' to call for backup from a physician staffed vehicle if in need of assistance? P9L15: Does it introduce selection bias, when prehospital clinician has to judge if hospital CPAP is available within 15 minutes?
--	--

	P9L23: Please specify what an oxygen alert card is. P9L26: Was reduced level of consciousness a contraindication? Please comment on this in the text. Or is it implicit in criteria 13) "unable to communicate"? P10L38: Please consider citing your protocol in this paragraph, since it describes allocation procedures in detail. P12L47: Consider stating the role of the supplier of the CPAP system and the role of the funder, even if they were not involved in study design, conduct or publication. P12L51: I am very much impressed with your public and patient involvement. There are many abbreviations in this paragraph though. P15L3: Figure 1: how come 66 patients were not enrolled because of "no equipment box on-board the attending ambulance"? Please comment on this. P16L3: There are two "table 1". P16L43: Two asterisks are noted with no following explanation? P18L10: Is it not 0% for major and $\leq 10\%$ for minor? (compare with table 1) P18L27: Is it possible to note the hospital diagnosis for this one patient that did not improve with CPAP? Could it be the one patient with a pulmonary embolism? P19L13: I wonder; is it a 100% adherence to allocation, when one patient in the "standard arm" had CPAP treatment because he had his own CPAP machine? P20L24: Please write out SAEs and clarify if there were no serious adverse events directly from the CPAP treatment. P21L32: What is the possible value range for this EQ-5D-5L scale? P21L39: No results are reported on post discharge resource use? P21L39: Are outputs from the planned cost-effectiveness evaluation of a definitive trial missing? P22L16: In my opinion, it is important to state here, that these two patients with pneumothorax did not receive CPAP treatment. P22L50: I agree. The problem is that a pneumothorax was not suspected, but perhaps in a physician back up EMS system, ultrasound would improve the ability to suspect or diagnose pneumothorax before CPAP application. P23L12: In our EMS organisation, 40 minutes on-scene time would
--	---

	be considered relatively long. Do you have the possibility to report on CPAP treatment times? Since this is a very interesting outcome to supplement on-scene times and conveyance times, when an ambulance service must decide upon implementing prehospital CPAP or not. P23L44: It might be worth to mention again in this paragraph, that recruitment rate could be raised with more trial trained staff. P24L16: Is it possible to elaborate why external validity to other settings is not clear? Perhaps the argument about prehospital physician staff for judging contraindications/eligibility criteria. P32L12: Block randomisation was used? The manuscript does not mention any block randomisation. P35L40: I am very impressed with your thorough preparations ahead of the study. P35L33, column 2: No mention of these patient reported adverse events from the 30-day follow-up questionnaire in the manuscript? P42L11: If possible, please add the date when follow-up was completed.
--	--

REVIEWER	FJAVIER BELDA University of Valencia, Spain
REVIEW RETURNED	16-Dec-2019

GENERAL COMMENTS	Pilot study to assess the feasibility of a large CT comparing prehospital CPAP and Standard O2 Therapy (SOT) on 30-day mortality in patients with ARF. Following the results the authors conclude that feasibility was not demonstrated. Limited compliance with CPAP, and difficulty in identifying patients who could benefit from CPAP, indicate that prehospital CPAP is unlikely to materially reduce mortality. This is an interesting study on an important intervention in ambulances in order to determine the effect on mortality of CPAP treatment. A study close to the perfection in design (already published) and granted by National Institute for Health Research's Health Technology Assessment . This pilot study is well justified in order to detect the feasibility of a new CT in a larger scale. The expected results according to the hypothesis may save lives and costs. Several issues must be considered by the authors in order to improve this manuscript. First, and of great importance, is the length of the manuscript and the content. It is excessively long and almost impossible to follow even for an interested reader. This precludes the understanding and reduces the relevance of the results.
---

	Please note the abundance of details that may be relevant for the design of the protocol but superfluous for this final report of the results. The Introduction is correct. However, for example in Methods, a 50% reduction should be desirable in: Participant enrolment and consent, Allocation concealment and blinding, Data collection and safety reporting and many other headlines. More than 90% reduction will be desirable in: Trial oversight, ethics and governance, Public and Patient Involvement The Results are too detailed preventing a clear understanding and this section should also be reduced. For example, there is an excessive detail of the effect of CPAP on individuals if one takes into account that is a pilot study (underpowered) and the results are less valuable (as is accepted by the authors: “we draw no conclusions from comparisons of outcomes between CPAP and standard care”). Discussion is again too broad. If you go to page 23/42 the summary of the discussion, it is read “...the challenges of providing prehospital CPAP, and characteristics of the patients who would benefit from CPAP”... This assertions are not clearly highlighted in the previous sentences because their excessive length. The reader is lost in the text. A second issue is the method of CPAP application itself. The authors consider CPAP as an all or nothing therapy while titration of applied CPAP is a key factor for its effect. Monitoring CPAP effect is thus very important for its titration and therefore for the therapeutic effect. It seems very strange the lack of efficacy of a therapy that clearly should reduce mortality when applied in the right way, time and place. Time to reach the hospital is another big issue. As the authors accept “The potential time advantage from prehospital administration of CPAP may be too small to produce meaningful benefit”. This implies that in a bigger trial, this should be considered a second major factor. Bias in the selection of the patients is a final consideration when only Finally in the Limitations section, something on impossibility of blinding the treatment should be added. Also, “Generalisability” may well be reduced to one to two sentences on the new CPAP device, as another limitation.
--	--

VERSION 1 – AUTHOR RESPONSE

REVIEWER ONE

We are very grateful for detailed review and have incorporated your suggestions to improve our manuscript:

- P1L3: I recommend no abbreviations in the title and suggest CPAP to be spelled out as Continuous Positive Airway Pressure. I also suggest including "versus standard oxygen therapy" to clarify in the title what both of the treatment arms of the RCT include.*
Thank you for this suggestion, we have edited the title accordingly.
- P2L18: Contributions: It might be simpler if each person's title e.g. "Chief Investigator" was noted only following the first time his or her name is mentioned. Then in the following*

paragraphs, only initials are used and no repetition of titles. Consider moving "The following drafted the manuscript" down to P3L2.

We have made this change as requested.

- *P4L13: Consider to delete specific centre name (according to BMJ Open submission guidelines, the name of a specific centre is irrelevant in the abstract).*
We have removed the name of the centre as suggested.
- *P4L32: It would be more precise if it was clarified in the abstract, that both the reported 30 day mortality and the following phrase are outcomes for the total study population. You may consider including results regarding any adverse events from CPAP rather than results from the post-hoc analysis of deceased patients, since the primary study objective concerns feasibility.*
We have added these details to the results section of the abstract.
- *P4L40: Consider concluding on the decision not to extend to a large-scale RCT in the abstract.*
We now include an additional sentence to highlight this point.
- *P4L44: Please specify the name of trial registry.*
We have added the name of the trial registry.
- *P4L46: BMJ Open prefers a standard sentence: "This work was supported by [name of funder] grant number [xxx]".*
We have edited the relevant sentence as requested.
- *P5L11: This section should not comment on the study's results but solely on the methods (please refer to BMJ Open submission guidelines for authors).*
We have edited this section as described above. Study results are no longer included.
- *P6L9: Is this the incidence rate in England?*
We have clarified this is the USA incidence rate. Definitive data are not available for England.
- *P6L25: Please specify the indications/diagnoses that CPAP is used for in hospital. P6L26: "prehospital diagnostics"?*
We have detailed that CPAP is used in UK hospitals to treat COPD, pneumonia, decompensated obstructive sleep apnoea, cardiogenic pulmonary oedema, and chest wall trauma
- *P7L19: An idea could be to cite your own protocol article here, in which you describe the EMS setting in detail. In your EMS setting, is it possible for 'ambulance clinicians' to call for backup from a physician staffed vehicle if in need of assistance?*
We have referenced the UK national guidance for ambulance service clinicians. Respectfully, we feel that this citation allows interested readers to study prehospital protocols if desired. In the UK setting it is unusual for a physician staffed vehicle to be called if in need of assistance. However some ambulance services have physician staffed helicopter emergency medical services, or voluntary physician response services. These are usually dispatched immediately after an emergency 999 call, rather than being called later for assistance. We have added a sentence to clarify this.

- P9L15: Does it introduce selection bias, when prehospital clinician has to judge if hospital CPAP is available within 15 minutes?*

We don't feel that this inclusion criterion will introduce selection bias as patients were randomised following study enrolment. However, we agree that this may influence generalisability of findings as now mentioned in the discussion.
- P9L23: Please specify what an oxygen alert card is.*

We have added a footnote to the table clarifying what an oxygen alert card is.
- P9L26: Was reduced level of consciousness a contraindication? Please comment on this in the text. Or is it implicit in criteria 13) "unable to communicate"?*

Thank you for highlighting this omission. Impaired level of consciousness was not an absolute contraindication to CPAP in line with British Thoracic Society guidelines.
- P10L38: Please consider citing your protocol in this paragraph, since it describes allocation procedures in detail.*

We have added the citation as requested.
- P12L47: Consider stating the role of the supplier of the CPAP system and the role of the funder, even if they were not involved in study design, conduct or publication.*

Thank you for this suggestion which have included.
- P12L51: I am very much impressed with your public and patient involvement. There are many abbreviations in this paragraph though.*

Thank you. This section has been heavily edited and now removes any undefined abbreviations.
- P15L3: Figure 1: how come 66 patients were not enrolled because of "no equipment box on-board the attending ambulance"? Please comment on this.*

We have added a footnote to the Figure to explain that on these 66 occasions ambulance service clinicians did not 'sign out' and carry a trial equipment box, and hence were unable to recruit a potentially eligible patient.
- P16L3: There are two "table 1".*

Thank you for highlighting this. We have made the necessary correction.
- P16L43: Two asterisks are noted with no following explanation?*

Thank you for highlighting this. We have removed this typographical error.
- P18L10: Is it not 0% for major and ≤10% for minor? (compare with table 1)*

That is correct; the targets were 0% for major and ≤10% for minor non-compliances. We think this is clear in the text.
- P18L27: Is it possible to note the hospital diagnosis for this one patient that did not improve with CPAP? Could it be the one patient with a pulmonary embolism?*

The patient who didn't improve had a primary diagnosis of pulmonary fibrosis.; we have added this detail as requested.
- P19L13: I wonder; is it a 100% adherence to allocation, when one patient in the "standard arm" had CPAP treatment because he had his own CPAP machine?*

Thank you for this observation. Technically the patient was compliant with the trial protocol as the trial intervention mask wasn't used. However, we agree this is slightly non-intuitive given that they did receive CPAP therapy (albeit with their own machine).

- *P20L24: Please write out SAEs and clarify if there were no serious adverse events directly from the CPAP treatment.*
We have now defined the serious adverse event (SAE) abbreviation and clarify that there were no serious adverse events directly from the CPAP treatment.
- *P21L32: What is the possible value range for this EQ-5D-5L scale?*
The range of the EQ-5D-5L scale is from – 0.285 to 1. We have provided a reference for this outcome.
- *P21L39: No results are reported on post discharge resource use?*
Thank you for highlighting this. We have now added a web appendix summarising the post discharge resource use.
- *P21L39: Are outputs from the planned cost-effectiveness evaluation of a definitive trial missing?*
Thank you for pointing this out. The cost-effectiveness evaluation is an extensive analysis that it is not possible to report within this paper. Separate publication is planned. We have added a sentence to clarify this.
- *P22L16: In my opinion, it is important to state here, that these two patients with pneumothorax did not receive CPAP treatment.*
We have now clarified this detail further.
- *P22L50: I agree. The problem is that a pneumothorax was not suspected, but perhaps in a physician back up EMS system, ultrasound would improve the ability to suspect or diagnose pneumothorax before CPAP application.*
Thank you, we agree, and have highlighted that our results might not be generalizable to more advanced EMS settings.
- *P23L12: In our EMS organisation, 40 minutes on-scene time would be considered relatively long. Do you have the possibility to report on CPAP treatment times? Since this is a very interesting outcome to supplement on-scene times and conveyance times, when an ambulance service must decide upon implementing prehospital CPAP or not.*
Thank you for this interesting point. We have added a sentence in the discussion to highlight this. Unfortunately we don't have information on CPAP treatment times available.
- *P23L44: It might be worth to mention again in this paragraph, that recruitment rate could be raised with more trial trained staff.*
Thank you for this suggestion. Unfortunately we don't have space to repeat this point within the limits of the journal word count.
- *P24L16: Is it possible to elaborate why external validity to other settings is not clear? Perhaps the argument about prehospital physician staff for judging contraindications/eligibility criteria.*
We have now added an additional sentence elaborating on why external validity may be limited to other settings.
- *P32L12: Block randomisation was used? The manuscript does not mention any block randomisation.*
Thank you very much for this astute observation. The protocol underwent an early amendment to simple randomisation before recruitment commenced. We have now clarified this in the methods. [Substantial amendment no.2 16Feb17 REC reference: 16/YH/0406]

- *P35L40: I am very impressed with your thorough preparations ahead of the study. Thank you very much.*
- *P35L33, column 2: No mention of these patient reported adverse events from the 30-day follow-up questionnaire in the manuscript?*
We have now included these results in a web appendix.
- *P42L11: If possible, please add the date when follow-up was completed.*
We have added this detail to the methods section.

REVIEWER TWO

Thank you for your review. We are gratified that the reviewer felt this to be an interesting, well conducted and important study.

- *A 50% reduction should be desirable in: Participant enrolment and consent, Allocation concealment and blinding, Data collection and safety reporting and many other headlines.*
We have edited these sections as requested.
- *More than 90% reduction will be desirable in: Trial oversight, ethics and governance, Public and Patient Involvement.*
We have edited these section accordingly to remove excess information.
- *No 95% confidence intervals are reported for risk differences in mortality and other outcomes between treatment groups.*
We have now included this information as requested in Table 5.
- *The Results are too detailed preventing a clear understanding and this section should also be reduced. For example, there is an excessive detail of the effect of CPAP on individuals if one takes into account that is a pilot study (underpowered) and the results are less valuable (as is accepted by the authors: “we draw no conclusions from comparisons of outcomes between CPAP and standard care”). Also, “Generalisability” may well be reduced to one to two sentences on the new CPAP device, as another limitation.*
Thank you for this viewpoint, which stands in contrast to Reviewer One who has requested additional details be added to each of these sections. We have edited the maunscript where possible, within the requests of Reviewer one.
- *A second issue is the method of CPAP application itself. The authors consider CPAP as an all or nothing therapy while titration of applied CPAP is a key factor for its effect. Monitoring CPAP effect is thus very important for its titration and therefore for the therapeutic effect. It seems very strange the lack of efficacy of a therapy that clearly should reduce mortality when applied in the right way, time and place.*
Thank you for this comment. We agree that CPAP is a titratable treatment, as delivered in the ACUTE trial. We have added an additional sentence to the methods to clarify this.
- *Time to reach the hospital is another big issue. As the authors accept “The potential time advantage from prehospital administration of CPAP may be too small to produce meaningful benefit”. This implies that in a bigger trial, this should be considered a second major factor.*
Thank you for this comment. We have added additional sentences to the discussion that highlight this point.

- *Finally in the Limitations section, something on impossibility of blinding the treatment should be added.*

Thank you for this interesting point. The ACUTE study was a pragmatic trial. Pragmatic RCTs are intended to evaluate the effects of interventions within routine medical care, and as such, do not typically mask treatment groups; moreover, pragmatic RCTs assess comparators that are available in routine medical practice, not masked placebos. With respect we therefore do not consider this issue a limitation.

VERSION 2 – REVIEW

REVIEWER	Vibe Maria Laden Nielsen, MD Centre for Prehospital and Emergency Research, Aalborg University & Aalborg University Hospital, Denmark
REVIEW RETURNED	01-May-2020

GENERAL COMMENTS	P6L21: Abstract, methods: consider stating the full eligibility criteria in the abstract, i.e. “respiratory distress AND peripheral oxygen saturations below BTS target levels DESPITE oxygen treatment” as it is described in the main text. The reader should be aware that the patients in this study are in more severe illness stages compared to many of the smaller previous studies in this research area (including the ones cited in your discussion), where patients were included if hypoxia was present before any treatment. This information would be preferable for the reader when interpreting your reported mortality numbers in the abstract. P6L35-45: “Limited compliance with CPAP” – I suppose this refers to CPAP being fully delivered in 74%, when target was set at 75%. Readers might be puzzled with the previous sentence “There were no major protocol violations/non-compliances.” Hence, the term “compliance” is used to describe two different feasibility results. I suggest it would be clearer with just “There were no major protocol violations.” P8L23: “This therapy is usually delivered by ambulance service personnel” (typo) P9L26: “last follow up completed” (typo) P13L5: “... and provided” (past tense) P13L17: “The choice of flow rate and oxygen delivery device was determined by ambulance clinicians” – what about level of FiO2? (In section “Setting and study population” above, you mention that your oxygen treatment is titrated). P18L3: This table is still incorrectly denoted “Table 1” in the pdf manuscript for proof reading. “Hospital ARF diagnosis”: n = 36 (CPAP) + n = 30 (SOT) does not equal 65 (total)? I suspect n in the SOT group was 29? Please spell out PE as pulmonary embolism in the footnote as the rest of the abbreviated diagnoses. No consistency in reporting decimals or not throughout the table. P22L36: Supplementary file 2: impressive and thorough documentation. P24L53: “... if these cases had been allocated to the intervention arm” – at P22L32 you write that one patient was in fact allocated to intervention arm, but did not receive CPAP. P25L3: You emphasise in your conclusion the “limited compliance with CPAP” – but in patients actually commencing the treatment, 81.6% continued until arrival at hospital or improvement. In my opinion, this fact deserves a little more discussion; firstly because the absolute difference is so small (only four patients who declined/did not tolerate the offered CPAP treatment). Secondly, it
--

	demonstrates that if the patient consents to CPAP treatment, patient compliance rate is acceptable and thirdly, because the result was not in line with compliance rates in previous CPAP studies with more ARF patients enrolled (previously provided references). P25L14: The potential CPAP treatment time was close to one hour (median on-scene times plus conveyance times). Other studies regarding prehospital CPAP for ARF with shorter CPAP treatment times have reported positive treatment effects, although using different clinical outcomes (e.g. citation 20 Hensel et al). P25L26: In my opinion, there is not necessarily a definite link between having a poor pre-morbid performance status or ceiling of treatment and the statement that CPAP might be futile – the role of treatment providers is also to provide symptom relief and not only to prevent death, which is difficult in patients that already have a high risk of mortality despite any treatment. Consider adding your result of change in patient VAS breathing score of -3 after CPAP treatment to the discussion. P25L52: Maybe it is worth to mention that collection of data on CPAP and other NIV treatments in the emergency department would be beneficial in any future trial, since this is a substantial confounder for outcomes in both groups. P26L53: In my opinion, there is still controversy about this treatment option and a large scale RCT would be of great value. Due to the abovementioned arguments, in my opinion, the compliance result is diverse. Respectfully, I find it a little unfair to state “feasibility was not demonstrated”. Consider emphasising your decision not to recommend a definitive trial on the lower than expected recruitment rate and challenge with selecting appropriate diagnoses that would benefit from CPAP. I congratulate you on finishing this manuscript of a most comprehensive trial; it has been an inspiration to review your work. P33L6: consider adding “up to 30 days after enrolment”.
--	---

REVIEWER	F. JAVIER BELDA Professor. Department of Surgery. University of Valencia, Spain
REVIEW RETURNED	01-May-2020

GENERAL COMMENTS	From my point of view the study is very good but it is a pity the authors have not found the best way to capture the attention and the interest of the reader. This version is even more detailed and longer that it was the first one. The length is against the paper in this occasion. A pragmatic trial is correct for reasearch but this does not prevent the possibility of detection and observer bias. If the authors do prefer not to mention this possibility is upon their responsibility.
---

VERSION 2 – AUTHOR RESPONSE

- *P6L21: Abstract, methods: consider stating the full eligibility criteria in the abstract, i.e. “respiratory distress AND peripheral oxygen saturations below BTS target levels DESPITE oxygen treatment” as it is described in the main text.*

Thank you for highlighting this. We have made the advised change.

- *“There were no major protocol violations/non-compliances.” Hence, the term “compliance” is used to describe two different feasibility results. I suggest it would be clearer with just “There were no major protocol violations.”*

Thanks – we have made the suggested edit to improve clarity.

- *P8L23: “This therapy is usually delivered by ambulance service personnel” (typo); P9L26: “last follow up completed” (typo); P13L5: “... and provided” (past tense)*

Thank you for finding these typos which have been corrected.

- *P13L17: “The choice of flow rate and oxygen delivery device was determined by ambulance clinicians” – what about level of FiO2? (In section “Setting and study population” above, you mention that your oxygen treatment is titrated).*

Thanks, we have clarified that therapy was titrated.

- *P18L3: This table is still incorrectly denoted “Table 1” in the pdf manuscript for proof reading. “Hospital ARF diagnosis”: $n = 36$ (CPAP) + $n = 30$ (SOT) does not equal 65 (total)? I suspect n in the SOT group was 29? Please spell out PE as pulmonary embolism in the footnote as the rest of the abbreviated diagnoses. No consistency in reporting decimals or not throughout the table.*

We have made all corrections as requested.

- *P24L53: “... if these cases had been allocated to the intervention arm” – at P22L32 you write that one patient was in fact allocated to intervention arm, but did not receive CPAP.*

Thank you for highlighting this discrepancy. We have now edited the sentence to avoid confusion.

- *P25L3: You emphasise in your conclusion the “limited compliance with CPAP” – but in patients actually commencing the treatment, 81.6% continued until arrival at hospital or improvement. In my opinion, this fact deserves a little more discussion; firstly because the absolute difference is so small (only four patients who declined/did not tolerate the offered CPAP treatment). Secondly, it demonstrates that if the patient consents to CPAP treatment, patient compliance rate is acceptable and thirdly, because the result was not in line with compliance rates in previous CPAP studies with more ARF patients enrolled (previously provided references).*

Thank you for this interesting viewpoint. We would respectfully suggest that overall compliance is the most relevant statistic, and that 81.6% compliance in patients who commenced CPAP is limited. We would therefore prefer not to revise this paragraph. However, we have added an additional sentence to the discussion to highlight that ACUTE results were not in line with compliance rates in previous CPAP studies with more ARF patients enrolled.

- *P25L14: The potential CPAP treatment time was close to one hour (median on-scene times plus conveyance times). Other studies regarding prehospital CPAP for ARF with shorter CPAP*

treatment times have reported positive treatment effects, although using different clinical outcomes (e.g. citation 20 Hensel et al).

Thank you for this point. We feel that we have covered this in the discussion 'generalisability' section:

".....international emergency medical services may have shorter on-scene times, use CPAP in less severe ARF, or use prehospital physicians with ultrasound skills to detect pneumothorax before CPAP applicationConsequently, efficacy could differ with other methods of delivering prehospital CPAP"

- *P25L26: In my opinion, there is not necessarily a definite link between having a poor pre-morbid performance status or ceiling of treatment and the statement that CPAP might be futile – the role of treatment providers is also to provide symptom relief and not only to prevent death, which is difficult in patients that already have a high risk of mortality despite any treatment. Consider adding your result of change in patient VAS breathing score of -3 after CPAP treatment to the discussion.*

Thank you for this important point. We have now added an additional sentence to the discussion.

- *P25L52: Maybe it is worth to mention that collection of data on CPAP and other NIV treatments in the emergency department would be beneficial in any future trial, since this is a substantial confounder for outcomes in both groups.*

Thanks, we have added this to the limitations section.

- *P26L53: In my opinion, there is still controversy about this treatment option and a large scale RCT would be of great value. Due to the abovementioned arguments, in my opinion, the compliance result is diverse. Respectfully, I find it a little unfair to state "feasibility was not demonstrated". Consider emphasising your decision not to recommend a definitive trial on the lower than expected recruitment rate and challenge with selecting appropriate diagnoses that would benefit from CPAP.*

We have now removed 'feasibility was not demonstrated' from the conclusions.

- *P33L6: consider adding "up to 30 days after enrolment".*

We have edited the table title as requested.